# Effect of a hospital-wide campaign on COVID-19 vaccination uptake among healthcare workers in the context of raised concerns for life-threatening side effects

Min Hyung Kim[1], Nak-Hoon Son[2,3], Yoon Soo Park[1], Ju Hyun Lee[4], Da Ae Kim[4], Yong Chan Kim[1]*

1 Division of Infectious Disease, Department of Internal Medicine, Yongin Severance Hospital, Yonsei University College of Medicine, Yongin-si, Gyeonggi-do, Republic of Korea, 2 Data Science Team (Biostatistician), Center for Digital Health, Yongin Severance Hospital, Yonsei University College of Medicine, Yongin-si, Gyeonggi-do, Republic of Korea, 3 Office of Biostatistics, Yongin Severance Hospital, Yonsei University College of Medicine, Yongin-si, Gyeonggi-do, Republic of Korea, 4 Infection Control Office, Yongin Severance Hospital, Yongin-si, Gyeonggi-do, Republic of Korea

☯ These authors contributed equally to this work.
* amomj@yuhs.ac

**Data Availability Statement:** All relevant data are within the paper and its Supporting Information files.

## Abstract

### Background

All healthcare workers (HCWs) in Yongin Severance Hospital were allocated to receive the ChAdOx1 nCov-19 vaccine according to national policy. A report of thrombosis and thrombocytopenia syndrome (TTS) associated with ChAdOx1 nCoV-19 led to hesitancy about receiving the second dose among HCWs who had received the first dose.

### Methods

From 7 to 14 May, 2021, we performed a survey to identify the factors associated with hesitancy about receiving the second vaccine dose among HCWs at the hospital who had received the first dose of the vaccine. Based on survey results, a hospital-wide campaign was implemented on 18 May 2021 to improve vaccine coverage. HCWs who completed the second dose completed a self-administered questionnaire to evaluate the effect of the campaign.

### Findings

Of 1,171 HCWs who had received the first dose of the vaccine, 71.5% completed the online survey, of whom 3.7% refused to take the second dose and 22.3% showed hesitancy. Hesitancy to receive a second dose was significantly associated with age under 30 years and concerns about TTS, and was less common among those who trusted effectiveness and safety of the vaccine. Among HCWs who received the first dose, 96.2% completed vaccination with the second dose between 27 May and 4 June, 2021. Of those who answered the

**Funding:** The author(s) received no specific funding for this work.

**Competing interests:** The authors have declared that no competing interests exist.

questionnaire asked about the timing of their decision to receive the second dose, 57.1% reported that they were motivated by the hospital-wide campaign.

## Conclusion

A tailored intervention strategy based on a survey can improve COVID-19 vaccination uptake among HCWs.

## Introduction

Globally, as of 2 July 2021, there were over 182 million confirmed cases of severe acute respiratory syndrome coronavirus 2 (SARS-CoV-2) infection, including 3.95 million deaths reported to the World Health Organization (WHO) [1]. In hopes of combating the disease by creating herd immunity, private and public institutions developed vaccines against SARS-CoV-2 at an unprecedented pace [2]. WHO launched COVID-19 Vaccines Global Access (COVAX) to ensure rapid development, manufacturing, and distribution of corona-virus disease (COVID-19) vaccine, which has helped accelerate the development of COVID-19 vaccines [3]. However, the accelerated course of vaccine development inevitably accompanies concerns for the potential side effects and complications, considering that vaccine development usually takes several years or even decades [4, 5]. Accordingly, news of serious side effects of new vaccines has raised concerns among many individuals and has become a major reason for vaccine hesitancy [6, 7]. Additionally, the emergence of variants without information on the efficacy of vaccines against these strains and a resurgence of COVID-19 cases worldwide despite increasing vaccination rates appear to have dampened desires for vaccination [8]. Overcoming public fear for the sake of herd immunity has become a major challenge for the worlds' leaders.

The Oxford-AstraZeneca adenovirus-vectored vaccine (ChAdOx1 nCoV-19) was the first COVID-19 vaccine authorised for use in the Republic of Korea on 10 February, 2021, and vaccination started on 26 February, 2021 [9]. Subsequently, a report of thrombosis and thrombocytopaenia syndrome (TTS) associated with the ChAdOx1 nCoV-19 vaccine was released from Europe on 9 April, 2021 [10], creating a lot of debate among experts over stopping vaccination with ChAdOx1 nCoV-19. On 12 April, 2021, the Korean guidelines for COVID-19 vaccination were changed due to reports of TTS among young adults who had received the ChAdOx1 nCov-19 vaccine. Considering the risks and benefits of vaccination, unvaccinated adults under 30 years were excluded from vaccination with the ChAdOx1 nCoV-19 vaccine. The Korean government also announced a policy to increase COVID-19 vaccination coverage to grant those who had been vaccinated an exemption from the 14-day mandatory quarantine that had been imposed on the close contacts of confirmed patients, and an exemption from mandatory mask wearing when outdoors [11].

In the Republic of Korea, healthcare workers (HCWs) were prioritised in vaccine allocation, and many HCWs were scheduled to receive the ChAdOx1 nCoV-19 vaccine. As of 22 March, 2021, the vaccine coverage ratio among the eligible Korean population was only 1.57% [12], largely due to constraints on vaccine supply and delivery around the world at that time [13]. We assumed that young HCWs had concerns about vaccination with ChAdOx1 nCoV-19, and that national policy alone would not be sufficient to address the issue. This study was conducted at a university affiliated hospital, where the percentage of individuals who completed the first dose of vaccination with the ChAdOx1 nCoV-19 vaccine was 63.3%. We conducted a

survey to evaluate HCWs' demands and to determine their intention regarding receiving a second dose of ChAdOx1 nCoV-19. Afterwards, we conducted a hospital-wide campaign based on the survey results to boost the vaccination rate of the second dose of ChAdOx1 nCoV-19 among HCWs.

## Methods

### Study population and study design

The Yongin Severance Hospital is a secondary care teaching hospital, with 708 beds, in the Republic of Korea. In accordance with the domestic policy for COVID-19 vaccination, all HCWs in the hospital were allocated to receive the ChAdOx1 nCov-19 vaccine, a replication-deficient adenoviral vector vaccine against COVID-19. The first dose of the vaccine was provided to HCWs from 8 to 19 March, 2021. Among a total of 1,851 HCWs, 1,171 (63.3%) received the vaccine during the period. After the release of the report of TTS associated with the ChAdOx1 nCoV-19 vaccine, hesitancy about accepting the second dose was observed among HCWs who had received the first dose. Measures were needed to resolve their concerns and to increase vaccination coverage of the second dose in HCWs.

### Survey

We prepared an online survey to evaluate HCWs' perception regarding the ChAdOx1 nCov-19 vaccine. The survey was administered to HCWs who had received the first dose of vaccine during 7–14 May, 2021. The survey consisted of questions that assessed their demographic characteristics, experience of adverse events after the first dose, COVID-19 experience and risk perception about COVID-19 severity, attitude regarding government recommended vaccinations and perception of COVID-19 vaccines, and their intention to accept the second dose of ChAdOx1 nCov-19 vaccine. Completing the questionnaire was voluntary, and participants were able to withdraw participation at any time. The questionnaire used in the research can be found in S1 Appendix.

### Hospital-wide campaigns

The results of the survey showed considerable hesitancy about receiving the second dose of ChAdOx1 nCov-19 vaccine among HCWs who are under the age of 30. Based on the results of the survey, a hospital-wide campaign was implemented from 18 May, 2021. The following measures were applied during the campaign: (1) the importance of COVID-19 vaccination was reemphasised through a large electronic display in the hospital lobby; (2) e-mail reminders were sent to HCWs to inform them that vaccination with the second dose was due, almost due, or past due date; (3) accurate information about ChAdOx1 nCov-19 vaccine was provided through education; and (4) a specialised clinical team for HCWs was created to respond promptly to any adverse events after vaccination. All HCWs who developed any symptoms after vaccination could visit the clinic at any time during working hours. The team paid special attention to severe adverse events, and were particularly alert to any cases of TTS. They checked the platelet count if a HCW developed symptoms suggestive of TTS, such as headache, dyspnoea, chest pain, and abdominal pain.

To evaluate the effect of the campaign, we conducted an additional survey for HCWs who completed the second dose. A simple, self-administered questionnaire asked about the timing of decision to receive the second dose. If participants answered 'decided after the hospital campaign', they were asked to select the reason why they decided to receive the second dose.

## Statistical analysis

Categorical variables are presented as frequencies and percentages and were compared using the Chi-square test or Fisher's exact test. Logistic regression was performed to identify predictive factors. With variables exhibiting significance in univariate analysis, as well as those with clinical relevance, we performed multivariate analysis. The validity of the variables was confirmed using the statistical variable selection method. All statistical analyses were performed using the R software version 4.0.2 (R Development Core Team, Vienna, Austria) and SAS software version 9.4 (SAS Institute Inc., Cary, NC, USA). Two-sided p-values $< 0.05$ were considered statistically significant.

## Ethics statement

This study was approved by the Institutional Review Board of Yonsei University Health System Clinical Trial Centre, and the study protocol adhered to the tenets of the Declaration of Helsinki. As the study was retrospective and the questionnaire was anonymous, the Institutional Review Board waived the requirement for written informed consent from the participants.

## Results

Fig 1 shows a process of vaccination, survey, and hospital-wide campaign and a flowchart of study population.

### Characteristics of healthcare workers who had received the first dose of the vaccine

Of 1,171 HCWs who had received the first dose of the vaccine, 837 (71.5%) completed the online survey. The characteristics of the respondents are summarised in Table 1. Of the respondents, 548 (65.5%) were women, and 514 (61.9%) were aged under 40 years. The most common adverse event reported was myalgia ($N = 601$, 71.8%), followed by injection site pain ($N = 585$, 69.9%), fatigue ($N = 582$, 69.5%), and fever ($N = 468$, 55.9%). Of the respondents, 206 (24.6%) reported that adverse events decreased their ability to work for several days. Most respondents did not have a history of COVID-19 and believed that if they contracted the disease, they were unlikely to develop severe disease. Overall, 746 (89.1%) respondents reported

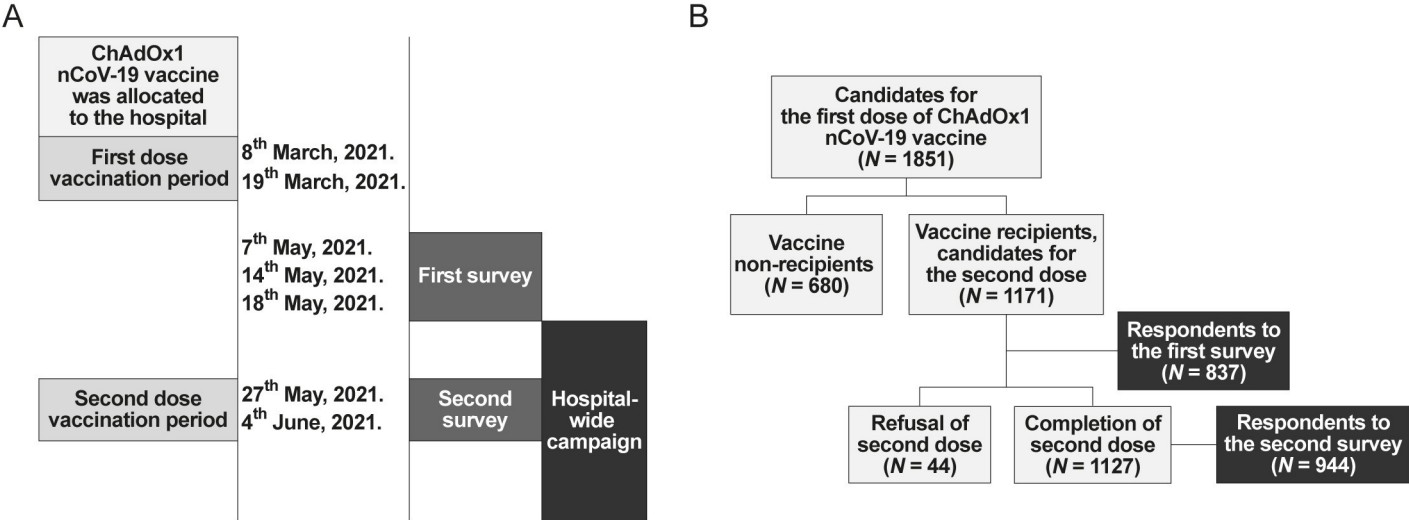

**Fig 1. Timeline of vaccination, survey, and the hospital-wide campaign.**

**Table 1. Characteristics of healthcare workers who had received the first dose of ChAdOx1 nCov-19 vaccine in the hospital.**

| Variables | All respondents (N = 837) | Intention to receive the second dose of ChAdOx1 nCoV-19 vaccine | | |
|---|---|---|---|---|
| | | Likely (N = 619) | Undecided or unwilling (N = 218) | P-value |
| **Demographic factors** | | | | |
| Gender | | | | < .001 |
| Female | 548 (65.5%) | 373 (60.3%) | 175 (80.3%) | |
| Male | 289 (34.5%) | 246 (39.7%) | 43 (19.7%) | |
| Age, years | | | | < .001 |
| Under 30 | 241 (29.0%) | 140 (22.8%) | 101 (46.5%) | |
| 30–39 | 273 (32.9%) | 192 (31.3%) | 81 (37.3%) | |
| 40–49 | 156 (18.8%) | 133 (21.7%) | 23 (10.6%) | |
| Over 50 | 161 (19.4%) | 149 (24.3%) | 12 (5.5%) | |
| Occupation | | | | < .001 |
| Nurse | 331 (39.6%) | 210 (33.9%) | 121 (55.5%) | |
| Doctor | 109 (13.0%) | 90 (14.5%) | 19 (8.7%) | |
| Others | 397 (47.4%) | 319 (51.5%) | 78 (35.8%) | |
| Smoking | | | | 0.01 |
| Yes, including former smokers | 93 (11.1%) | 79 (12.8%) | 14 (6.4%) | |
| No | 744 (88.9%) | 540 (87.2%) | 204 (93.6%) | |
| Alcohol | | | | 0.439 |
| Yes | 422 (50.4%) | 317 (51.2%) | 105 (48.1%) | |
| No | 415 (49.6%) | 302 (48.8%) | 113 (51.8%) | |
| Comorbidities | | | | 0.279 |
| Yes | 167 (20.0%) | 129 (20.8%) | 38 (17.4%) | |
| No | 670 (80.1%) | 490 (79.2%) | 180 (82.6%) | |
| Children in household | | | | < .001 |
| Yes | 360 (43.0%) | 301 (48.6%) | 59 (27.1%) | |
| No | 477 (57.0%) | 318 (51.4%) | 159 (72.9%) | |
| Parents in household | | | | 0.119 |
| Yes | 279 (33.3%) | 197 (31.8%) | 82 (37.6%) | |
| No | 558 (66.7%) | 422 (68.2%) | 136 (62.4%) | |
| **Experience of adverse event to the first dose** | | | | |
| Fever | | | | < .001 |
| Yes | 468 (55.9%) | 310 (50.1%) | 158 (72.5%) | |
| No | 369 (44.1%) | 309 (49.9%) | 60 (27.5%) | |
| Vomiting | | | | < .001 |
| Yes | 25 (3.0%) | 8 (1.3%) | 17 (7.8%) | |
| No | 812 (97.0%) | 611 (98.7%) | 201 (92.2%) | |
| Diarrhoea | | | | < .001 |
| Yes | 47 (5.6%) | 26 (4.2%) | 21 (9.6%) | |
| No | 790 (94.4%) | 593 (95.8%) | 197 (90.4%) | |
| Headache | | | | < .001 |
| Yes | 416 (49.7%) | 278 (44.9%) | 138 (63.3%) | |
| No | 421 (50.3%) | 341 (55.1%) | 80 (36.7%) | |
| Fatigue | | | | < .001 |
| Yes | 582 (69.5%) | 398 (64.3%) | 184 (84.4%) | |
| No | 255 (30.5%) | 221 (35.7%) | 34 (15.6%) | |
| Chill | | | | < .001 |

*(Continued)*

**Table 1.** (*Continued*)

| Variables | All respondents (N = 837) | Intention to receive the second dose of ChAdOx1 nCoV-19 vaccine | | |
| --- | --- | --- | --- | --- |
| | | Likely (N = 619) | Undecided or unwilling (N = 218) | P-value |
| Yes | 450 (53.8%) | 296 (47.8%) | 154 (70.6%) | |
| No | 387 (46.2%) | 323 (52.2%) | 64 (29.4%) | |
| Myalgia | | | | < .001 |
| Yes | 601 (71.8%) | 419 (67.7%) | 182 (83.5%) | |
| No | 236 (28.2%) | 200 (32.3%) | 36 (16.5%) | |
| Arthralgia | | | | < .001 |
| Yes | 221 (26.4%) | 137 (22.1%) | 84 (38.5%) | |
| No | 616 (73.6%) | 482 (77.9%) | 134 (61.5%) | |
| Others | | | | 0.013[a] |
| Yes | 14 (1.7%) | 6 (1.0%) | 8 (3.7%) | |
| No | 823 (98.3%) | 613 (99.0%) | 210 (96.3%) | |
| Injection site pain | | | | < .001 |
| Yes | 585 (69.9%) | 408 (65.9%) | 177 (81.2%) | |
| No | 252 (30.1%) | 211 (34.1%) | 41 (18.8%) | |
| Injection site redness | | | | < .001 |
| Yes | 160 (19.1%) | 99 (16.0%) | 61 (28.0%) | |
| No | 677 (80.9%) | 520 (84.0%) | 157 (72.0%) | |
| Injection site swelling | | | | < .001 |
| Yes | 187 (22.3%) | 120 (19.4%) | 67 (30.7%) | |
| No | 650 (77.7%) | 499 (80.6%) | 151 (69.3%) | |
| Decrease in work efficiency due to adverse event | | | | < .001 |
| Yes | 206 (24.6%) | 107 (17.3%) | 99 (45.4%) | |
| No | 631 (75.4%) | 512 (82.7%) | 119 (54.6%) | |
| **COVID-19 experience and risk perception about COVID-19 severity** | | | | |
| Did you experience with COVID-19 symptoms without confirmed diagnosis? | | | | 0.029 |
| Yes | 127 (15.2%) | 84 (13.6%) | 43 (19.7%) | |
| No | 710 (84.8%) | 535 (86.4%) | 175 (80.3%) | |
| Did you have a test for COVID-19 ever? | | | | 0.081 |
| Yes | 180 (21.5%) | 124 (20.0%) | 56 (25.7%) | |
| No | 657 (78.5%) | 495 (80.0%) | 162 (74.3%) | |
| Do you know someone who had been confirmed with COVID-19? | | | | 0.106 |
| Yes | 101 (12.1%) | 68 (11.0%) | 33 (15.1%) | |
| No | 736 (87.9%) | 551 (89.0%) | 185 (84.9%) | |
| How do you think you are likely to be when infected to SARS-CoV-2? | | | | 0.032 |
| Severe | 147 (17.6%) | 99 (16.0%) | 48 (22.0%) | |
| Moderate | 522 (62.4%) | 385 (62.2%) | 137 (62.8%) | |
| Mild | 168 (20.1%) | 135 (21.8%) | 33 (15.1%) | |
| **Variables related to vaccination with the second dose** | | | | |
| Previous compliance with recommended vaccination | | | | 0.009 |
| Always | 746 (89.1%) | 562 (90.8%) | 184 (84.4%) | |
| Sometimes or never | 91 (10.9%) | 57 (9.2%) | 34 (15.6%) | |
| Do you think that ChAdOx1 nCoV-19 vaccine is effective? | | | | < .001 |
| Yes | 749 (89.5%) | 588 (95.0%) | 161 (73.9%) | |
| No | 88 (10.5%) | 31 (5.0%) | 57 (26.2%) | |

(*Continued*)

**Table 1.** (Continued)

| Variables | All respondents (N = 837) | Intention to receive the second dose of ChAdOx1 nCoV-19 vaccine | | |
|---|---|---|---|---|
| | | Likely (N = 619) | Undecided or unwilling (N = 218) | P-value |
| Do you think that ChAdOx1 nCoV-19 vaccine is safe? | | | | < .001 |
| Yes | 436 (52.1%) | 405 (65.4%) | 31 (14.2%) | |
| No | 401 (47.9%) | 214 (34.6%) | 187 (85.8%) | |
| Do you have concerns about the vaccine induced thrombotic thrombocytopenia? | | | | < .001 |
| Yes | 643 (76.8%) | 429 (69.3%) | 214 (98.2%) | |
| No | 194 (23.2%) | 190 (30.7%) | 4 (1.8%) | |
| Perceived prevalence of the vaccine induced thrombotic thrombocytopenia | | | | 0.002 |
| <1/1,000,000 | 427 (51.0%) | 338 (54.6%) | 89 (40.8%) | |
| 1/100,000~1/1,000,000 | 287 (34.3%) | 194 (31.3%) | 93 (42.7%) | |
| >1/100,000 | 123 (14.7%) | 87 (14.1%) | 36 (16.5%) | |

COVID-19, coronavirus disease 2019; SARS-CoV-2, severe acute respiratory syndrome coronavirus 2.

[a]Fisher's Exact Test.

good compliance with recommended vaccinations in the past. Although 749 (89.5%) answered that they considered the ChAdOx1 nCov-19 vaccine to be effective, about half of the respondents (N = 401, 47.9%) reported that they did not consider the vaccine to be safe, and 643 (76.8%) had concerns about TTS related with the ChAdOx1 nCov-19 vaccine, with 123 (14.7%) overestimating the prevalence of TTS. Of the respondents, 619 (74.0%) reported that they intended to receive a second dose of the vaccine, while 218 (26.0%) reported that they were hesitant or intended to refuse the second dose of the vaccine.

## Hesitancy regarding the second dose of the vaccine

Of the respondents, 187 (22.3%) reported hesitancy about receiving the second dose of the ChAdOx1 nCov-19 vaccine (Table 2). In the univariate analysis, hesitancy was associated with female sex; age <40 years; being a nurse, non-smoker, or childless; having experienced an adverse event after receiving the first dose; perceived decreased work ability attributed to vaccination; knowing someone confirmed with COVID-19; perceived severity of the illness if infected with SARS-CoV-2; lower compliance with recommended vaccination in the past; distrust of the effectiveness or safety of the ChAdOx1 nCov-19 vaccine; and concerns about TTS.

Multivariate analysis showed a strong association ($P < 0.001$) between hesitancy and some variables. Respondents under 30 years of age were more likely to hesitate to receive a second dose than those over 50 years (adjusted odds ratio [aOR]: 5.8, 95% CI: 2.21–15.23, $P < 0.001$). In contrast, respondents who trusted the effectiveness (aOR: 0.3, 95% CI: 0.17–0.51, $P < 0.001$) and safety (aOR: 0.27, 95% CI: 0.16–0.43, $P < 0.001$) were less likely to hesitate to receive the second dose compared with those who distrusted the vaccine. Concerns about the ChAdOx1 nCov-19 vaccine-induced TTS were significantly associated with hesitancy (aOR: 7.54, 95% CI: 2.44–23.25, $P < 0.001$).

## Effect of hospital-wide campaign

Among 1,171 HCWs who received the first dose of the vaccine, 1,127 (96.2%) completed vaccination with the second dose during the week of 27 May to 4 June, 2021. During this period,

**Table 2. Factors associated with hesitancy to the second dose of ChAdOx1 nCov-19 vaccine (N = 187).**

| Variables | Univariate analysis | | | Multivariable analysis | | |
|---|---|---|---|---|---|---|
| | OR | 95% CI | p value | OR | 95% CI | p value |
| **Demographic factors** | | | | | | |
| Gender | | | | | | |
| Female | 2.97 | (1.98, 4.45) | < .001 | 1.57 | (0.89, 2.77) | 0.121 |
| Male | REF | | | REF | | |
| Age, years | | | | | | |
| Under 30 | 8.42 | (4.32, 16.42) | < .001 | 5.8 | (2.21, 15.23) | < .001 |
| 30–39 | 4.73 | (2.41, 9.26) | < .001 | 3.52 | (1.48, 8.37) | 0.004 |
| 40–49 | 2.14 | (0.99, 4.6) | 0.052 | 1.56 | (0.65, 3.78) | 0.320 |
| Over 50 | REF | | | REF | | |
| Occupation | | | | | | |
| Nurse | 2.22 | (1.56, 3.16) | < .001 | 1.09 | (0.69, 1.74) | 0.715 |
| Doctor | 0.87 | (0.49, 1.56) | 0.647 | 1.01 | (0.48, 2.09) | 0.987 |
| Others | REF | | | REF | | |
| Smoking | | | | | | |
| Yes, including former smokers | 0.43 | (0.22, 0.82) | 0.011 | 0.81 | (0.34, 1.92) | 0.626 |
| No | REF | | | REF | | |
| Alcohol | | | | | | |
| Yes | 0.98 | (0.71, 1.36) | 0.922 | | | |
| No | REF | | | | | |
| Comorbidities | | | | | | |
| Yes | 0.81 | (0.53, 1.24) | 0.340 | | | |
| No | REF | | | | | |
| Children in household | | | | | | |
| Yes | 0.39 | (0.27, 0.55) | < .001 | 1.87 | (1.01, 3.45) | 0.047 |
| No | REF | | | REF | | |
| Parents in household | | | | | | |
| Yes | 1.2 | (0.85, 1.69) | 0.307 | | | |
| No | REF | | | REF | | |
| **Experience of adverse event to the first dose** | | | | | | |
| Fever | | | | | | |
| Yes | 2.66 | (1.86, 3.8) | < .001 | 0.91 | (0.54, 1.54) | 0.733 |
| No | REF | | | REF | | |
| Vomiting | | | | | | |
| Yes | 6.66 | (2.78, 15.96) | < .001 | 2.99 | (0.96, 9.32) | 0.059 |
| No | REF | | | REF | | |
| Diarrhoea | | | | | | |
| Yes | 2.58 | (1.39, 4.78) | 0.003 | 1.42 | (0.65, 3.07) | 0.380 |
| No | REF | | | REF | | |
| Headache | | | | | | |
| Yes | 2.1 | (1.5, 2.94) | < .001 | 0.68 | (0.41, 1.11) | 0.122 |
| No | REF | | | REF | | |
| Fatigue | | | | | | |
| Yes | 3.03 | (1.97, 4.64) | < .001 | 1.54 | (0.84, 2.81) | 0.161 |
| No | REF | | | REF | | |
| Chill | | | | | | |
| Yes | 2.69 | (1.89, 3.83) | < .001 | 1.17 | (0.69, 1.99) | 0.570 |

(*Continued*)

**Table 2.** (Continued)

| Variables | Univariate analysis | | | Multivariable analysis | | |
|---|---|---|---|---|---|---|
| | OR | 95% CI | p value | OR | 95% CI | p value |
| No | REF | | | REF | | |
| Myalgia | | | | | | |
| Yes | 2.31 | (1.53, 3.51) | < .001 | 1.14 | (0.63, 2.06) | 0.669 |
| No | REF | | | REF | | |
| Arthralgia | | | | | | |
| Yes | 2.11 | (1.48, 2.99) | < .001 | 1.04 | (0.65, 1.67) | 0.863 |
| No | REF | | | REF | | |
| Others | | | | | | |
| Yes | 3.39 | (1.08, 10.63) | 0.037 | 1.25 | (0.35, 4.49) | 0.728 |
| No | REF | | | REF | | |
| Injection site pain | | | | | | |
| Yes | 2.51 | (1.65, 3.79) | < .001 | 1.06 | (0.62, 1.81) | 0.841 |
| No | REF | | | REF | | |
| Injection site redness | | | | | | |
| Yes | 1.97 | (1.34, 2.9) | 0.001 | 0.93 | (0.54, 1.63) | 0.810 |
| No | REF | | | REF | | |
| Injection site swelling | | | | | | |
| Yes | 1.82 | (1.26, 2.64) | 0.001 | 1.06 | (0.62, 1.81) | 0.833 |
| No | REF | | | REF | | |
| Decrease in work efficiency due to adverse event | | | | | | |
| Yes | 3.99 | (2.8, 5.69) | < .001 | 1.98 | (1.25, 3.14) | 0.004 |
| No | REF | | | REF | | |
| **COVID-19 experience and risk perception about COVID-19 severity** | | | | | | |
| Did you experience with COVID-19 symptoms without confirmed diagnosis? | | | | | | |
| Yes | 1.47 | (0.95, 2.26) | 0.083 | | | |
| No | REF | | | | | |
| Did you have a test for COVID-19 ever? | | | | | | |
| Yes | 1.34 | (0.91, 1.97) | 0.136 | | | |
| No | REF | | | | | |
| Do you know someone who had been confirmed with COVID-19? | | | | | | |
| Yes | 1.61 | (1.02, 2.55) | 0.042 | 1.32 | (0.74, 2.33) | 0.349 |
| No | REF | | | REF | | |
| How do you think you are likely to be when infected to SARS-CoV-2? | | | | | | |
| Severe | 1.97 | (1.13, 3.43) | 0.017 | 1.45 | (0.72, 2.9) | 0.298 |
| Moderate | 1.57 | (0.99, 2.49) | 0.055 | 1.57 | (0.89, 2.77) | 0.116 |
| Mild | REF | | | REF | | |
| **Variables related to vaccination with the second dose** | | | | | | |
| Previous compliance with recommended vaccination | | | | | | |
| Always | 0.55 | (0.34, 0.89) | 0.016 | 0.49 | (0.26, 0.9) | 0.021 |
| Sometimes or never | REF | | | REF | | |
| Do you think that ChAdOx1 nCoV-19 vaccine is effective? | | | | | | |
| Yes | 0.17 | (0.1, 0.27) | < .001 | 0.3 | (0.17, 0.54) | < .001 |
| No | REF | | | REF | | |
| Do you think that ChAdOx1 nCoV-19 vaccine is safe? | | | | | | |
| Yes | 0.11 | (0.07, 0.16) | < .001 | 0.27 | (0.16, 0.43) | < .001 |
| No | REF | | | REF | | |

(*Continued*)

**Table 2.** (Continued)

| Variables | Univariate analysis | | | Multivariable analysis | | |
|---|---|---|---|---|---|---|
| | OR | 95% CI | p value | OR | 95% CI | p value |
| Do you have concerns about the vaccine induced thrombotic thrombocytopenia? | | | | | | |
| Yes | 20.26 | (7.42, 55.36) | < .001 | 7.54 | (2.44, 23.25) | < .001 |
| No | REF | | | REF | | |
| Perceived prevalence of the vaccine induced thrombotic thrombocytopenia | | | | | | |
| <1/1,000,000 | 0.73 | (0.44, 1.19) | 0.202 | | | |
| 1/100,000~1/1,000,000 | 1.28 | (0.78, 2.11) | 0.330 | | | |
| >1/100,000 | REF | | | | | |

COVID-19, coronavirus disease 2019; SARS-CoV-2, severe acute respiratory syndrome coronavirus 2.

944 (83.8%) HCWs answered the questionnaire about the timing of the decision to receive the second dose of the vaccine (Fig 2). Overall, 220 respondents (23.3%) reported that they had decided to receive the second dose after the hospital-wide campaign. Among them, 125 (56.8%) selected 'hospital-wide campaign' as the motive for their decision. The second most common reason was 'national policy' ($N$ = 46, 20.9%), followed by 'recommendation by others' ($N$ = 31, 14.1%) and 'positive information obtained from media' ($N$ = 17, 7.7%).

## Discussion

COVID-19 vaccination of HCWs is important to provide herd immunity in hospitals and to reduce the risk of nosocomial transmission of SARS-CoV-2 [14]. Even though vaccines may change over time due to newly emerging issues [15, 16], documenting efforts to boost immunization in the context of constraints on vaccine supply and delivery is meaningful. In this study, although all HCWs in the hospital were designated to receive the ChAdOx1 nCov-19 vaccine, the coverage with the first dose was low. Furthermore, in the survey assessing the intention to receive the second dose, 22.3% of respondents were hesitant to receive the second dose and 3.7% of respondents expressed outright refusal. We evaluated factors associated with vaccine hesitancy, and based on the results, a hospital-wide campaign was implemented to increase vaccination coverage of the second dose. After the campaign, the rate of vaccination uptake of the second dose was 96.2% among HCWs who had received the first dose. The survey conducted among HCWs after their second vaccination revealed that 57.1% of those who decided to receive a second dose after the hospital-wide campaign, were motivated by the campaign.

Vaccination rate drops when the perceived side effects of a vaccine outweigh the disease severity [17]. We demonstrated that female sex and young age were risk factors associated with hesitancy to receive the second dose of ChAdOx1 nCov-19 vaccine in univariate analysis. Since initial reports have shown that life-threatening TTS has occurred mainly in young-aged women, persons with one of these factors would likely be hesitant to get vaccinated with a second dose [18]. Other factors such as reduced work efficiency due to adverse events after the first dose and previous poor compliance with other vaccines were also related to vaccine hesitancy, in accordance with previous studies [19]. Compared to previous reports on vaccine hesitancy [6, 16, 20, 21], concerns about life-threatening side effects were very common among respondents. Therefore, efforts were needed to resolve and address the concerns identified in the survey. In addition to offer correct information about TTS, a clinic for vaccinated HCWs was established to manage adverse events after vaccination.

The second survey revealed that 76.7% of participants had decided to accept the second dose of the vaccine before the campaign began. Previous reports on HCWs in France also

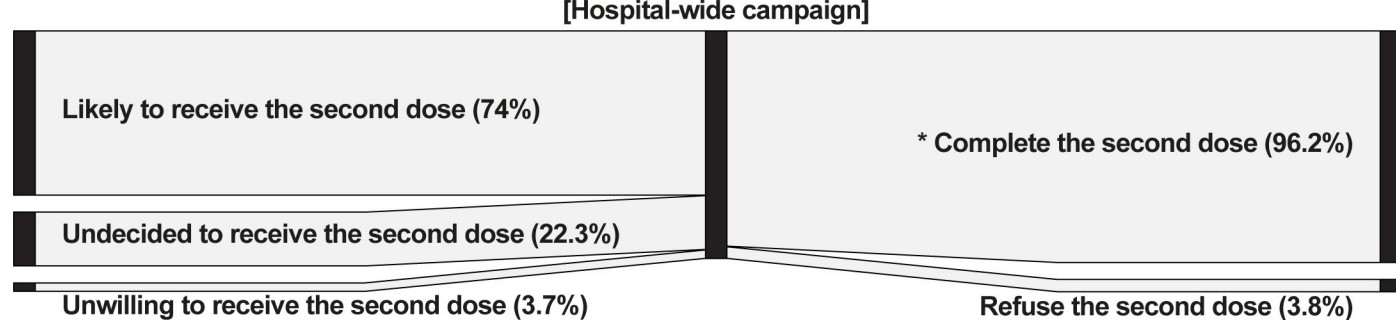

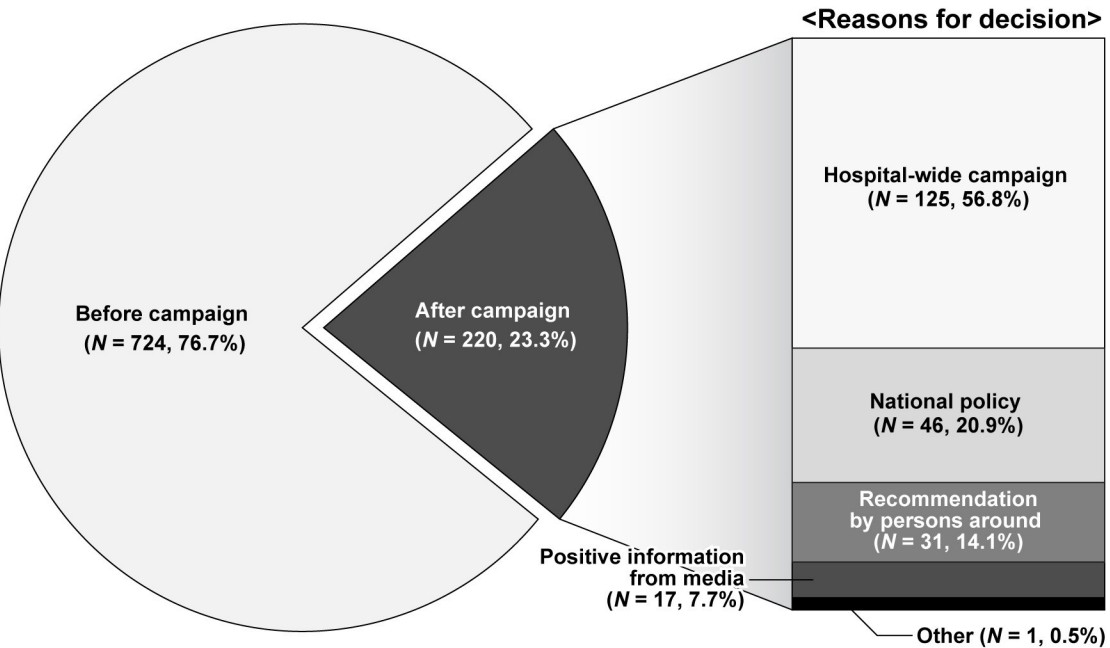

**Fig 2. Responses to when the second dose recipients decided to get vaccinated and reasons why the respondents decided to receive the second dose after the hospital-wide campaign.**

suggested a similar percentage of vaccine acceptance [6]. In contrast to the general population [22], stronger intentions to receive vaccination have already been established among HCWs regardless of their job type. Responsibility as HCWs and high risk of exposure to COVID-19 might contribute to a favourable attitude towards the vaccine. This reflects the social consensus formed among HCWs confronting an unprecedented pandemic. Therefore, the HCWs of this study might be easily convinced of the necessity of vaccination and lower vaccine hesitancy.

Authentic information and dedicating resources to managing potential adverse events are important motivators for vaccination. Previous reports have revealed that misinformation often leads to vaccine hesitancy, which prompts the need for education of the public [7, 17, 23]. The importance of education targeting HCWs has been emphasized in studies of other vaccines [24]. We strived to correct misinformation related to ChAdOx1 nCov-19 vaccine through education. Several opportunities were provided to HCWs to be educated during a hospital-wide campaign. Such campaigns should be tailored to meet the needs of the public to address the ongoing

challenge [25]: For example, cases of TTS were reported among persons who had received the ChAdOx1 nCov-19 vaccine, after which many HCWs who had been vaccinated with first dose expressed concerns about vaccination with the second dose. Therefore, our campaign focused on addressing concerns related to TTS. In addition to providing accurate information about TTS, we developed a specialized clinical team so that HCWs can receive medical consultation promptly whenever they complained of any symptoms after the second dose. Most responders said that the hospital campaign was a major determinant in their decision to accept the second dose of vaccination. Although it was not determined which component of the campaign had the greatest effects on resolving the hesitancy for the second dose, we believe that education and a specialized clinical team for HCWs played a major role in addressing vaccine hesitancy. Our findings provide insight for the direction of national policies to improve vaccine coverage in the general population, especially among young adults who agonize over vaccination due to an imbalance in the associated risks and benefits of vaccination [17]. Further study evaluating the effects of tailored campaign on vaccine uptake in the general population is warranted.

The second major reason for the decision to accept the second dose of vaccine was the incentives provided according to national policy for individuals who completed the scheduled vaccination. Other researchers have also described the importance of incentives for frontline workers [14, 26], and this study confirmed that incentives can provide an inducement for vaccination. Judicious employment of incentives is required, taking into account the fatigue that people might have felt since the pandemic began, and their hope of returning to "normalcy" in the context of the pandemic.

This study has several limitations. Due to the inherent limitation of a survey-based study, we could not conduct a direct comparison between the responders to the first and second surveys. However, it is reasonable to assume that the responders are a representative sample of the hospital staff with a response rate of > 70%. Second, the factors that influenced the participants who made early decisions regarding vaccination, as well as those who refused to receive the first dose of the vaccine, could not be determined in this study. This study focused on individuals who were likely to be affected by news on TTS adverse events and highlighted measures to improve second dose uptake amid heightened alerts for vaccine side effects. In order to take measures to improve overall vaccination rates in other settings, reasons for outright refusal of ChAdOx1 nCov-19 vaccination need to be investigated. Lastly, our results should be interpreted with caution because the surveys were conducted at a single centre; hence, the findings might not be generalisable. Further study is warranted to evaluate the effects of interventions to promote COVID-19 vaccination in other settings.

## Conclusions

This is the first study showing the effect of a hospital-wide campaign to promote COVID-19 vaccination among HCWs. For successful campaigning, it is necessary to plan tailored intervention strategies considering the characteristics of HCWs. A hospital-wide campaign based on a survey of HCWs (emphasising the importance of vaccination, providing accurate information about vaccines, sending reminders for vaccination using email, and reducing concerns about adverse events through a specialised clinic team for HCWs) contributed to improve COVID-19 vaccination coverage. Our findings need to be emphasized when scaling up vaccination coverage for the general population.

## Supporting information

**S1 Appendix. The questionnaire used in the study.**
(DOCX)

**S2 Appendix. Raw datasets used in the analysis.**
(XLSX)

## Acknowledgments

We thank the hospital staff at Yongin Severance Hospital. We thank Medical Illustration & Design, part of the Medical Research Support Services of Yonsei University College of Medicine, for all artistic support related to this work.

## Author Contributions

**Conceptualization:** Yong Chan Kim.

**Data curation:** Min Hyung Kim, Nak-Hoon Son, Ju Hyun Lee, Yong Chan Kim.

**Formal analysis:** Min Hyung Kim, Nak-Hoon Son, Yong Chan Kim.

**Investigation:** Min Hyung Kim, Nak-Hoon Son, Yong Chan Kim.

**Methodology:** Min Hyung Kim, Yong Chan Kim.

**Resources:** Nak-Hoon Son, Ju Hyun Lee, Da Ae Kim, Yong Chan Kim.

**Supervision:** Yoon Soo Park.

**Validation:** Yoon Soo Park, Ju Hyun Lee, Da Ae Kim, Yong Chan Kim.

**Writing – original draft:** Min Hyung Kim, Nak-Hoon Son, Yong Chan Kim.

**Writing – review & editing:** Min Hyung Kim, Yong Chan Kim.

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
