## [Decision Letter · Decision Letter 0]

20 Aug 2021

PONE-D-21-22598

Effect of a hospital-wide campaign on the COVID-19 vaccination uptake among healthcare workers

PLOS ONE

Dear Dr. Kim,

Thank you for submitting your manuscript to PLOS ONE. After careful consideration, we feel that it has merit but does not fully meet PLOS ONE’s publication criteria as it currently stands. Therefore, we invite you to submit a revised version of the manuscript that addresses the points raised during the review process.

Please consider the survey design carefully, as one of the reviewers suggested.

We look forward to receiving your revised manuscript.

Kind regards,

Etsuro Ito

Academic Editor

PLOS ONE

Journal Requirements:

2. Please state whether complete of the questionnaire was voluntary and participants were able to withdraw participation at any time.

Reviewers' comments:

Reviewer's Responses to Questions

**Comments to the Author**

1. Is the manuscript technically sound, and do the data support the conclusions?

Reviewer #1: Partly

Reviewer #2: Yes

2. Has the statistical analysis been performed appropriately and rigorously? 

Reviewer #1: No

Reviewer #2: No

3. Have the authors made all data underlying the findings in their manuscript fully available?

Reviewer #1: Yes

Reviewer #2: No

4. Is the manuscript presented in an intelligible fashion and written in standard English?

Reviewer #1: No

Reviewer #2: Yes

5. Review Comments to the Author

Reviewer #1: Review Report

Manuscript Number: PONE-D-21-22598

Title: Effect of a hospital-wide campaign on the COVID-19 vaccination uptake among healthcare workers

Background:

This study focusses on the vaccine hesitancy of hospital workers in Republic of Korean context. It shows that hospital-wide campaign to motivate health workers to get vaccinated – lowered vaccine hesitancy among health workers in getting the second dose.

Comments:

While I enjoyed reading the paper, I have some major reservations.

First, I am not sure about the novelty of the study. It is well known that ‘directed campaign’ and ‘generating awareness on vaccination’, encourage people to get vaccinated. Such campaign is also seen as an important means to mitigate the adverse affect of vaccine misinformation. Therefore, while I acknowledge that this might be the first study on hospital workers on vaccine hesitancy, the results are not surprising. I would suggest the authors to focus more on the contribution of the study/ paper. In other words, what is the incremental leaning from this study? In this context, I would like to highlight that it should be easier to convince hospital workers (compared to general public) about the benefits of vaccination and lower their vaccine hesitancy.

Second, I have concerns about the survey design. As mentioned in the study, 1,171 (60.3%) of the total hospital workers (1,851) received vaccination during the period of 8 to 19 March, 2021. The survey (of this study) focusses on 1,171 health workers who received the first dose of vaccination. To me, it is more important to focus on the health workers who did not want to get vaccinated (even the first dose). Because, those who got the first dose, know that without the second dose their vaccine will not be effective. Therefore, it is easier to manage the vaccine hesitancy challenge of this group. But the real challenge lies with the unvaccinated health workers (36.7% of total health workers!). I would suggest the authors to focus on this group as well – and explore what is the cause of vaccine hesitancy for this group (who did not even want to get the first dose).

Third, based on the univariate and multivariate analysis, the study has identified a few factors that contribute to vaccine hesitancy (female workers, younger workers…). However, no compelling arguments are presented by the authors as to why these factors lead to a higher level of vaccine hesitancy. As the general management literature suggests, female workers are more rule abiding, more compassionate and care more about society. It is not clear why female workers would be more hesitant. Similarly, it is not clear why younger workers would be more hesitant (may be, they have lower covid risk?). These points are important – because the hospital wide campaign should be linked to such factors.

Fourth, (somewhat linked to my previous point), there is no link discussed between the hospital wide campaign and the factors contributing to vaccine hesitancy. It looks like that any general campaign could work. This view questions the relevance of this study.

Reviewer #2: The authors report on a study on hesitancy to get vaccinated with the AstraZeneca vaccine among healthcare workers (HCW) in Yonin Severence Hospital in South Korea. HCW participated in online surveys after the first vaccine dose and were interviewed about vaccine hesitancy to receive a second doses. After the survey, a hospital-wide information campaign was launched to promote compliance with the second dose. After the second dose a second online survey was conducted among HCW to investigate the reasons for compliance with the second dose.

The topic is definitely relevant in the context of vaccination hesitancy, particularly in the context of the ongoing discussion of mandatory vaccines. Naturally, such studies have their limitations. The manuscript is well written and easy to understand. I think it has merit and should be published. However, I have a number of comments and suggestions that should definitely be addressed. I believe my comments can be easily addressed.

Kristan Schneider

Comments:

General comments:

1. Reasons to get vaccinated change rapidly throughout the pandemic. While some people did not want to get vaccinated immediately, their concerns naturally diminished over time, as more people were already vaccinated, aggressive variants are spreading, and incentives to get vaccinated are being implemented. This should be discussed a bit more.

2. Throughout the ms, the vaccine is called ChAdOx1 nCoV-19. It should be mentioned that it Is called also AZD 1222 or the AstraZeneca vaccine. I believe this will improve accessibility to a broader audience.

3. It should be discussed that the campaign can presumably not be scaled up to the full extend. It might work well in hospitals, but running such information campaigns in the general population is different, particularly point (4). However, some parts of the campaign are scalable. In so far this study gives insights in what could be done in the general population to promote the vaccine. I think the ms would profit from a broader discussion.

Major comments:

1. I have major concerns regarding the statistical analyses. Looking at the numbers in Table 1, I did not believe the p-vales reported. Hence, I ran some tests in R and obtained totally different p-values (which seem plausible to me). For instance, smoking: I ran the following R-code:

Fishers exact test:

fisher.test(matrix(c(79,540,14,204),ncol=2,nrow=2))

and obtained a p-value of 0.01171

Chi-square test with Yates correction:

chisq.test(matrix(c(79,540,14,204),ncol=2,nrow=2))

p-value: 0.01483

Chi-square test without Yates correction:

chisq.test(matrix(c(79,540,14,204),ncol=2,nrow=2),correct = FALSE)

p-value: 0.01042

The reported p-value was <0.001

It is similar for all other variables I checked. The p-values reported just do not seem plausible to me, and I obtained different p-values for all variables I checked.

I believe it is necessary to revise the table and the main text accordingly.

2. Table 1: Please be more specific on the statistical test. In all 2x2 tables a fisher’s exact should be performed. If a chi-square test is performed, it is unclear whether the Yates correction was applied (with this sample size I would not do it as it tends to over-correct).

3. Page 12: It needs to be specified more clearly what models were run. I suppose the univariate analyses means a logistic regression just with an intercept and this variable. With the multivariate analysis on the other hand all variables were forced into the logistic model.

I have objections against both. By the univariate analyses, a lot of models are fit, all of them presumably with a poor fit. However, since there is just one covariate and the intercept, the estimated regression coefficient will always appear significantly different form 0. This is actually seen by the small p-values. By the multivariate analysis, the model tends to overfit. After correction for multiple testing hardly any odd ratio is significantly different from 1.

An appropriate approach is to do a model selection based on the AIC criterium (in R function stepAIC – you specify the minimal model which is just and the maximum model). It will return the optimal model based on the AIC. The p-values of the z-tests for the regression coefficients that remain in the model should then be corrected for multiple testing by the Holm-method (I do not recommend the Hochberg method in this case).

In any case the model fit needs to be reported in terms of AIC and Null-Deviance and Deviance. The logistic regression is only justified if the model properly fits.

4. Introduction/Methods (L90-97): other reasons for vaccine hesitancy could have been the fact that the former “South African” variant spread and the AstraZeneca is not properly protecting from that variant. Another reason for hesitancy for the second dose might have been, that at this time it was reported that a longer vaccination schedule would improve the vaccine’s effectiveness. This should be discussed.

Minor comments:

L81: vaccine coverage rate: what is the vaccine coverage rate? Is it really a rate (with unit 1/time) or is it the percentage of vaccinated individuals in a certain group?

L81: I am confused about the 1.19%. This cannot be the percentage in the hospital where 63.3% got vaccinated. Is the 1.19% a figure for all of Korea? Also, it is important to report accessibility of the vaccine in this context. Vaccine distribution was characterized by bottlenecks in February and March this year.

L83: What is the completion rate? Is it the fraction of vaccinated individuals that complete the vaccination schedule? Again this would not be a rate (it is a dimensionless quantity, rates have unit 1/time).

L125-129: the statistical methods should be described in more detail, particularly whether the Yates correction was used with the chi-square tests, or what routines were run. Was it proc glm in SAS or glm() in R? etc.

Trivial comments:

L138: study -> the study

P12L6: sSARS -> SARS

6. PLOS authors have the option to publish the peer review history of their article (what does this mean?). If published, this will include your full peer review and any attached files.

Reviewer #1: No

Reviewer #2: No

---

## [Author Response · Author response to Decision Letter 0]

13 Sep 2021

To the reviewers

Thank you for your detailed evaluation of our article. We have corrected the text based on your recommendations. Corrections are shown in red and our responses to your comments are provided below.

Reviewer #1 

While I enjoyed reading the paper, I have some major reservations.

1. First, I am not sure about the novelty of the study. It is well known that ‘directed campaign’ and ‘generating awareness on vaccination’, encourage people to get vaccinated. Such campaign is also seen as an important means to mitigate the adverse affect of vaccine misinformation. Therefore, while I acknowledge that this might be the first study on hospital workers on vaccine hesitancy, the results are not surprising. I would suggest the authors to focus more on the contribution of the study/ paper. In other words, what is the incremental leaning from this study? In this context, I would like to highlight that it should be easier to convince hospital workers (compared to general public) about the benefits of vaccination and lower their vaccine hesitancy.

Response: Thank you for your opinion. We revised our article to put an emphasis on the fact that our survey was conducted in the context of raised concerns for adverse events of the vaccine and to highlight the role of our campaign providing immediate medical services to promote vaccine uptake. Furthermore, we decided to include additional discussion regarding the implication of our research for the general population. We revised our script as follows: 

In the Discussion section on page 19, lines 220-244: 

“The second survey revealed that 76.7% of participants had decided to accept the second dose of the vaccine before the campaign began. Previous reports on HCWs in France also suggested a similar percentage of vaccine acceptance. In contrast to the general population [1], stronger intentions to receive vaccination have already been established among HCWs regardless of their job type. Responsibility as HCWs and high risk of exposure to COVID-19 might contribute to a favourable attitude towards the vaccine. This reflects the social consensus formed among HCWs confronting an unprecedented pandemic. Therefore, the HCWs of this study might be easily convinced of the necessity of vaccination and lower vaccine hesitancy.

Authentic information and dedicating resources to managing potential adverse events are important motivators for vaccination. Previous reports have revealed that misinformation often leads to vaccine hesitancy, which prompts the need for education of the public. The importance of education targeting HCWs has been emphasized in studies of other vaccines. We strived to correct misinformation related to ChAdOx1 nCov-19 vaccine through education. Several opportunities were provided to HCWs to be educated during a hospital-wide campaign. Such campaigns should be tailored to meet the needs of the public to address the ongoing challenge: For example, cases of TTS were reported among persons who had received the ChAdOx1 nCov-19 vaccine, after which many HCWs who had been vaccinated with first dose expressed concerns about vaccination with the second dose. Therefore, our campaign focused on addressing concerns related to TTS. In addition to providing accurate information about TTS, we developed a specialized clinical team so that HCWs can receive medical consultation promptly whenever they complained of any symptoms after the second dose. Most responders said that the hospital campaign was a major determinant in their decision to accept the second dose of vaccination. Although it was not determined which component of the campaign had the greatest effects on resolving the hesitancy for the second dose, we believe that education and a specialized clinical team for HCWs played a major role in addressing vaccine hesitancy. Our findings provide insight for the direction of national policies to improve vaccine coverage in the general population, especially among young adults who agonize over vaccination due to an imbalance in the associated risks and benefits of vaccination [2]. Further study evaluating the effects of tailored campaign on vaccine uptake in the general population is warranted.” 

2. Second, I have concerns about the survey design. As mentioned in the study, 1,171 (60.3%) of the total hospital workers (1,851) received vaccination during the period of 8 to 19 March, 2021. The survey (of this study) focusses on 1,171 health workers who received the first dose of vaccination. To me, it is more important to focus on the health workers who did not want to get vaccinated (even the first dose). Because, those who got the first dose, know that without the second dose their vaccine will not be effective. Therefore, it is easier to manage the vaccine hesitancy challenge of this group. But the real challenge lies with the unvaccinated health workers (36.7% of total health workers!). I would suggest the authors to focus on this group as well – and explore what is the cause of vaccine hesitancy for this group (who did not even want to get the first dose).

Response: We totally agree with your opinion. It is important to explore causes of vaccine refusal for the first dose, but we couldn’t pay attention to them. Instead, we focused on hesitancy for the second dose and highlighted measures to improve second dose uptake because concerns for a second dose peaked among HCWs who received the first dose with reports of unexpected severe adverse events (TTS) associated with ChAdOx1 nCov-19. Therefore, taking that into account, we expounded on your comments as a limitation as follows: 

In the Discussion section on page 20, lines 253-259

“Second, the factors that influenced the participants who made early decisions regarding vaccination, as well as those who refused to receive the first dose of the vaccine, could not be determined in this study. This study focused on individuals who were likely to be affected by news on TTS adverse events and highlighted measures to improve second dose uptake amid heightened alerts for vaccine side effects. In order to take measures to improve overall vaccination rates in other settings, reasons for outright refusal of ChAdOx1 nCov-19 vaccination need to be investigated.”

3. Third, based on the univariate and multivariate analysis, the study has identified a few factors that contribute to vaccine hesitancy (female workers, younger workers…). However, no compelling arguments are presented by the authors as to why these factors lead to a higher level of vaccine hesitancy. As the general management literature suggests, female workers are more rule abiding, more compassionate and care more about society. It is not clear why female workers would be more hesitant. Similarly, it is not clear why younger workers would be more hesitant (may be, they have lower covid risk?). These points are important – because the hospital wide campaign should be linked to such factors.

Response: Thank you for pointing this out. We agree with you and included the following explanations: 

In the Discussion section on page 18, lines 210-214:

“Vaccination rate drops when the perceived side effects of a vaccine outweigh the disease severity. We demonstrated that female sex and young age were risk factors associated with hesitancy to receive the second dose of ChAdOx1 nCov-19 vaccine in univariate analysis. Since initial reports have shown that life-threatening TTS has occurred mainly in young-aged women, persons with one of these factors would likely be hesitant to get vaccinated with a second dose [3]. “ 

4. Fourth, (somewhat linked to my previous point), there is no link discussed between the hospital wide campaign and the factors contributing to vaccine hesitancy. It looks like that any general campaign could work. This view questions the relevance of this study.

Response: Thank you for your comment. We believe that education targeting HCWs and a specialized clinical team for HCWs to respond promptly to any adverse events after vaccination have played a major role in addressing vaccine hesitancy, and we have already mentioned this in the Discussion section. We further emphasized the effects of these and revised our article as follows: 

In the Discussion section on page 19, lines 227-244: 

“Authentic information and dedicating resources to managing potential adverse events are important motivators for vaccination. Previous reports have revealed that misinformation often leads to vaccine hesitancy, which prompts the need for education of the public. The importance of education targeting HCWs has been emphasized in studies of other vaccines. We strived to correct misinformation related to ChAdOx1 nCov-19 vaccine through education. Several opportunities were provided to HCWs to be educated during a hospital-wide campaign. Such campaigns should be tailored to meet the needs of the public to address the ongoing challenge: For example, cases of TTS were reported among persons who had received the ChAdOx1 nCov-19 vaccine, after which many HCWs who had been vaccinated with first dose expressed concerns about vaccination with the second dose. Therefore, our campaign focused on addressing concerns related to TTS. In addition to providing accurate information about TTS, we developed a specialized clinical team so that HCWs can receive medical consultation promptly whenever they complained of any symptoms after the second dose. Most responders said that the hospital campaign was a major determinant in their decision to accept the second dose of vaccination. Although it was not determined which component of the campaign had the greatest effects on resolving the hesitancy for the second dose, we believe that education and a specialized clinical team for HCWs played a major role in addressing vaccine hesitancy. Our findings provide insight for the direction of national policies to improve vaccine coverage in the general population, especially among young adults who agonize over vaccination due to an imbalance in the associated risks and benefits of vaccination [2]. Further study evaluating the effects of tailored campaign on vaccine uptake in the general population is warranted.” 

Reviewer #2: 

The authors report on a study on hesitancy to get vaccinated with the AstraZeneca vaccine among healthcare workers (HCW) in Yonin Severence Hospital in South Korea. HCW participated in online surveys after the first vaccine dose and were interviewed about vaccine hesitancy to receive a second doses. After the survey, a hospital-wide information campaign was launched to promote compliance with the second dose. After the second dose a second online survey was conducted among HCW to investigate the reasons for compliance with the second dose.

The topic is definitely relevant in the context of vaccination hesitancy, particularly in the context of the ongoing discussion of mandatory vaccines. Naturally, such studies have their limitations. The manuscript is well written and easy to understand. I think it has merit and should be published. However, I have a number of comments and suggestions that should definitely be addressed. I believe my comments can be easily addressed.

General comments:

1. Reasons to get vaccinated change rapidly throughout the pandemic. While some people did not want to get vaccinated immediately, their concerns naturally diminished over time, as more people were already vaccinated, aggressive variants are spreading, and incentives to get vaccinated are being implemented. This should be discussed a bit more.

Response: Thank you for your detailed comment. 

1. We agree to your opinion that concerns about vaccination would diminish naturally over time. Some of HCWs decided to get vaccinated with second dose regardless of our campaign, as shown in results of this study. However, our study highlights the effects of our campaign on responders who decided to receive a second dose after the hospital-wide campaign. Indeed, 57.1% of those responded that the hospital campaign was a major determinant in their decision to accept the second dose of vaccination. 

2. We agree with your opinion regarding that vaccine incentives would have influenced on improving vaccine coverage. This has already been described in discussion section, on page 19, line 241. 

3. Emergence of variants would likely to have influenced to get vaccinated with ChAdOx1 nCoV-19 due to reduced vaccine efficacy compared with mRNA vaccines manufactured by Pfizer or Moderna. However, it is difficult that variants could have influenced on vaccination with second dose in our study since variants were rarely observed in South Korea at the time this survey was conducted.

2. Throughout the ms, the vaccine is called ChAdOx1 nCoV-19. It should be mentioned that it Is called also AZD 1222 or the AstraZeneca vaccine. I believe this will improve accessibility to a broader audience.

Response: We agree with your comment, so we added the official name of the vaccine, which is Oxford-AstraZeneca adenovirus-vectored vaccine, at its first appearance.

In the Introduction section on page 3, line 71: 

“The Oxford-AstraZeneca adenovirus-vectored vaccine (ChAdOx1 nCoV-19)” 

3. It should be discussed that the campaign can presumably not be scaled up to the full extend. It might work well in hospitals, but running such information campaigns in the general population is different, particularly point (4). However, some parts of the campaign are scalable. In so far this study gives insights in what could be done in the general population to promote the vaccine. I think the ms would profit from a broader discussion.

Response: We agree with your comment, so, we supplemented our article by adding discussion for the implication for general population. We revised discussion accordingly. 

In the discussion section page 19, line 220-244: 

“The second survey revealed that 76.7% of participants had decided to accept the second dose of the vaccine before the campaign began. Previous reports on HCWs in France also suggested a similar percentage of vaccine acceptance. In contrast to the general population [1], stronger intentions to receive vaccination have already been established among HCWs regardless of their job type. Responsibility as HCWs and high risk of exposure to COVID-19 might contribute to a favourable attitude towards the vaccine. This reflects the social consensus formed among HCWs confronting an unprecedented pandemic. Therefore, the HCWs of this study might be easily convinced of the necessity of vaccination and lower vaccine hesitancy.

Authentic information and dedicating resources to managing potential adverse events are important motivators for vaccination. Previous reports have revealed that misinformation often leads to vaccine hesitancy, which prompts the need for education of the public. The importance of education targeting HCWs has been emphasized in studies of other vaccines. We strived to correct misinformation related to ChAdOx1 nCov-19 vaccine through education. Several opportunities were provided to HCWs to be educated during a hospital-wide campaign. Such campaigns should be tailored to meet the needs of the public to address the ongoing challenge: For example, cases of TTS were reported among persons who had received the ChAdOx1 nCov-19 vaccine, after which many HCWs who had been vaccinated with first dose expressed concerns about vaccination with the second dose. Therefore, our campaign focused on addressing concerns related to TTS. In addition to providing accurate information about TTS, we developed a specialized clinical team so that HCWs can receive medical consultation promptly whenever they complained of any symptoms after the second dose. Most responders said that the hospital campaign was a major determinant in their decision to accept the second dose of vaccination. Although it was not determined which component of the campaign had the greatest effects on resolving the hesitancy for the second dose, we believe that education and a specialized clinical team for HCWs played a major role in addressing vaccine hesitancy. Our findings provide insight for the direction of national policies to improve vaccine coverage in the general population, especially among young adults who agonize over vaccination due to an imbalance in the associated risks and benefits of vaccination [2]. Further study evaluating the effects of tailored campaign on vaccine uptake in the general population is warranted.”

Major comments:

1. I have major concerns regarding the statistical analyses. Looking at the numbers in Table 1, I did not believe the p-vales reported. Hence, I ran some tests in R and obtained totally different p-values (which seem plausible to me). For instance, smoking: I ran the following R-code:

Fishers exact test:

fisher.test(matrix(c(79,540,14,204),ncol=2,nrow=2))

and obtained a p-value of 0.01171

Chi-square test with Yates correction:

chisq.test(matrix(c(79,540,14,204),ncol=2,nrow=2))

p-value: 0.01483

Chi-square test without Yates correction:

chisq.test(matrix(c(79,540,14,204),ncol=2,nrow=2),correct = FALSE)

p-value: 0.01042

The reported p-value was <0.001

It is similar for all other variables I checked. The p-values reported just do not seem plausible to me, and I obtained different p-values for all variables I checked.

I believe it is necessary to revise the table and the main text accordingly.

Response: Thank you for your comment. To make sure we had analyzed the data correctly, we double checked the result about smoking status using SAS. We reproduced p-value 0.0104, which was exactly the same figure as you had calculated with Chi-square test without Yates using program R. The results were also the same with the other two methods, Chi-square with Yates correction and Fishers exact test. We presented p-value by rounding it up to the decimal point 0.01. The result makes no difference whether you use Yates correction or not. I am afraid that you confused the number with that of the above. We present our results below. 

Capture of the Table1 ‘Smoking’

2. Table 1: Please be more specific on the statistical test. In all 2x2 tables a fisher’s exact should be performed. If a chi-square test is performed, it is unclear whether the Yates correction was applied (with this sample size I would not do it as it tends to over-correct).

Response: According to a study by Ronald Fisher (1922), a Chi-squared test (or better yet, a G-test) can be used when the sample size is big. However, it only provides approximation because the distribution of the samples that is calculated is an approximation which is equal to the theoretical Chi-squared distribution. The approximation is invalid when the sample sizes are small or data are unequally distributed across the table, resulting in low counts on the cells predicated on a null hypothesis (the expected counts). The usual rule of thumb is that chi-squared test is not suitable when the expected values in any of the cells of a contingency table are below 5, or below 10 when there is only one degree of freedom [3]. In our study, we annotated with lower case ‘a’ when Fisher’s exact test is required. We present the results below. 

Capture of the Table 1 ‘Others’ and annotation

3. Page 12: It needs to be specified more clearly what models were run. I suppose the univariate analyses means a logistic regression just with an intercept and this variable. With the multivariate analysis on the other hand all variables were forced into the logistic model.

I have objections against both. By the univariate analyses, a lot of models are fit, all of them presumably with a poor fit. However, since there is just one covariate and the intercept, the estimated regression coefficient will always appear significantly different form 0. This is actually seen by the small p-values. By the multivariate analysis, the model tends to overfit. After correction for multiple testing hardly any odd ratio is significantly different from 1.

An appropriate approach is to do a model selection based on the AIC criterium (in R function stepAIC – you specify the minimal model which is just and the maximum model). It will return the optimal model based on the AIC. The p-values of the z-tests for the regression coefficients that remain in the model should then be corrected for multiple testing by the Holm-method (I do not recommend the Hochberg method in this case).

In any case the model fit needs to be reported in terms of AIC and Null-Deviance and Deviance. The logistic regression is only justified if the model properly fits.

Response: We appreciate your comment. Here are our answers. First of all, the purpose of the second analysis was to figure out the factors that leads to hesitancy. We carried out logistic regression analysis as described in “Statistical analysis” section, confirming the significance of each variable with univariate analysis. With the variables of p-value 0.05 or below by global test as well as the ones with clinical relevance, we performed multivariate analysis. Validity of the variables was confirmed by statistical variable selection method. Generally, researchers use Akaike Information Criterion (AIC) or Bayesian Information Criterion (BIC) to find the fittest model among various ones the methods present. Furthermore, AIC and BIC is useful to prevent reckless addition of variables for the purpose of gaining fitness. In this sense, we assume using them to check the adjusted effect of the variables is inappropriate. 

4. Introduction/Methods (L90-97): other reasons for vaccine hesitancy could have been the fact that the former “South African” variant spread and the AstraZeneca is not properly protecting from that variant. Another reason for hesitancy for the second dose might have been, that at this time it was reported that a longer vaccination schedule would improve the vaccine’s effectiveness. This should be discussed.

Response: Thank you for your comments. It is true that the efficacy of the ChAdOx1 nCov-19 vaccine against variants is lower than that of mRNA vaccine. However, at the time this survey was conducted, variants were rare in South Korea and were observed in only a few foreign entrants from overseas. Therefore, I think that the emergence of variants is less likely to influence hesitancy for second dose. Also, since the schedule of second dose was set to 11-12 weeks after the first vaccine (recommended longest time to be extended) in our institution, it is difficult for HCWs to hesitant to second dose to improve the efficacy of vaccine.

Minor comments:

1. L81: vaccine coverage rate: what is the vaccine coverage rate? Is it really a rate (with unit 1/time) or is it the percentage of vaccinated individuals in a certain group?

Response: Thank you for your detailed comment. We decided to use ‘ratio’ instead of ‘rate’. We changed our article as follows. 

In the Introduction section on page 3, line 84-85: 

“As of 22 March, 2021, the vaccine coverage ratio among eligible Korean population was only 1.57%, largely due to constraints on vaccine supply and delivery around the world at that time [4].”

2. L81: I am confused about the 1.19%. This cannot be the percentage in the hospital where 63.3% got vaccinated. Is the 1.19% a figure for all of Korea? Also, it is important to report accessibility of the vaccine in this context. Vaccine distribution was characterized by bottlenecks in February and March this year.

Response: The number 1.19% was produced with vaccine eligible Korean population as denominator. We corrected the miscalculation and revised our article accordingly. 

In the Introduction section on page 3, line 84-85: 

“As of 22 March, 2021, the vaccine coverage ratio among eligible Korean population was only 1.57%, largely due to constraints on vaccine supply and delivery around the world at that time [4].”

3. L83: What is the completion rate? Is it the fraction of vaccinated individuals that complete the vaccination schedule? Again this would not be a rate (it is a dimensionless quantity, rates have unit 1/time).

Response: We appreciate your detailed comment. We changed it to ‘percentage of completion’. 

4. L125-129: the statistical methods should be described in more detail, particularly whether the Yates correction was used with the chi-square tests, or what routines were run. Was it proc glm in SAS or glm() in R? etc.

Response: Generally, Yates correction is used only in 2 by 2 case. In this study, we performed Chi-square test or Fisher’s exact test for 2 by r case not only for 2 by 2 case. As a results, we did not consider Yates correction into account. However, we would like to inform you that we performed Chi-square test or Fisher’s exact test with precision. Furthermore, we used proc logstic and proc freq procedure of SAS and R to produce figure. We revised our article accordingly. 

In the subsection ‘Statistical analysis’ in Method section on page 5, line 131-137: 

“Categorical variables are presented as frequencies and percentages and were compared using the Chi-square test or Fisher’s exact test. Logistic regression was performed to identify predictive factors. With variables exhibiting significance in univariate analysis, as well as those with clinical relevance, we performed multivariate analysis. The validity of the variables was confirmed using the statistical variable selection method. All statistical analyses were performed using the R software version 4.0.2 (R Development Core Team, Vienna, Austria) and SAS software version 9.4 (SAS Institute Inc., Cary, NC, USA). Two-sided p-values < 0.05 were considered statistically significant.”

Trivial comments:

L138: study -> the study

Response: thank you for pointing out our mistake. We corrected it. 

P12L6: sSARS -> SARS

Response: thank you for pointing out our mistake. We corrected it.

---

## [Decision Letter · Decision Letter 1]

22 Sep 2021

Effect of a hospital-wide campaign on COVID-19 vaccination uptake among healthcare workers in the context of raised concerns for life-threatening side effects

PONE-D-21-22598R1

Dear Dr. Kim,

We’re pleased to inform you that your manuscript has been judged scientifically suitable for publication and will be formally accepted for publication once it meets all outstanding technical requirements.

Kind regards,

Etsuro Ito

Academic Editor

PLOS ONE

Reviewers' comments:

Reviewer's Responses to Questions

**Comments to the Author**

1. If the authors have adequately addressed your comments raised in a previous round of review and you feel that this manuscript is now acceptable for publication, you may indicate that here to bypass the “Comments to the Author” section, enter your conflict of interest statement in the “Confidential to Editor” section, and submit your "Accept" recommendation.

Reviewer #1: All comments have been addressed

2. Is the manuscript technically sound, and do the data support the conclusions?

Reviewer #1: Yes

3. Has the statistical analysis been performed appropriately and rigorously? 

Reviewer #1: Yes

4. Have the authors made all data underlying the findings in their manuscript fully available?

Reviewer #1: Yes

5. Is the manuscript presented in an intelligible fashion and written in standard English?

Reviewer #1: Yes

6. Review Comments to the Author

Reviewer #1: The article has improved significantly; I understand that some comments cannot be addressed directly and the authors have discussed such issues as limitations. I appreciate it.

7. PLOS authors have the option to publish the peer review history of their article (what does this mean?). If published, this will include your full peer review and any attached files.

Reviewer #1: No

---

## [Editor Report · Acceptance letter]

24 Sep 2021

PONE-D-21-22598R1 

Effect of a hospital-wide campaign on COVID-19 vaccination uptake among healthcare workers in the context of raised concerns for life-threatening side effects 

Dear Dr. Kim:

I'm pleased to inform you that your manuscript has been deemed suitable for publication in PLOS ONE. Congratulations! Your manuscript is now with our production department. 

Kind regards, 

on behalf of

Prof. Etsuro Ito 

Academic Editor

PLOS ONE